# Discovery of Salidroside as a Novel Non-Coding RNA Modulator to Delay Cellular Senescence and Promote BK-Dependent Apoptosis in Cerebrovascular Smooth Muscle Cells of Simulated Microgravity Rats

**DOI:** 10.3390/ijms241914531

**Published:** 2023-09-26

**Authors:** Yiling Ge, Bin Zhang, Jibo Song, Qinglin Cao, Yingrui Bu, Peijie Li, Yungang Bai, Changbin Yang, Manjiang Xie

**Affiliations:** 1Department of Aerospace Physiology, Key Laboratory of Aerospace Medicine of Ministry of Education, Fourth Military Medical University, Xi’an 710032, China; 15339046217@163.com (Y.G.); fmmuzhangbin@yeah.net (B.Z.); sssongjibo@163.com (J.S.); caoqinglin@126.com (Q.C.); b99961008411@163.com (Y.B.); lpj6467@163.com (P.L.); baiyun_1123@163.com (Y.B.); 2Military Medical Innovation Center, Fourth Military Medical University, Xi’an 710032, China

**Keywords:** microgravity, non-coding RNAs, salidroside, cellular senescence, BK-dependent apoptosis

## Abstract

Cardiovascular aging has been reported to accelerate in spaceflights, which is a great potential risk to astronauts’ health and performance. However, current exercise routines are not sufficient to reverse the adverse effects of microgravity exposure. Recently, salidroside (SAL), a valuable medicinal herb, has been demonstrated to display an important role for prevention and treatment in cardiovascular and other diseases. In the present work, Sprague–Dawley rats with four-week tail-suspension hindlimb-unloading were used to simulate microgravity effects on the cardiovascular system. We found that intragastrical administration of SAL not only significantly decreased the expressions of senescence biomarkers, such as P65 and P16, but also obviously increased the expressions of BK-dependent apoptotic genes, including the large-conductance calcium-activated K^+^ channel (BK), Bax, Bcl-2, and cleaved caspase-3, in vascular smooth muscle cells (VSMCs) in vivo and in vitro. In addition, relative non-coding RNAs were screened, and a luciferase assay identified that SAL increased apoptosis by activating LncRNA-FLORPAR, inhibiting miR-193, and then triggering the activity of the BK-α subunit. Our work indicated that SAL is a novel non-coding RNA modulator for regulating the LncRNA-FLORPAR sponging miR-193 pathway, which significantly promoted BK-dependent apoptosis and delayed cerebrovascular aging-like remodeling during simulated microgravity exposure. Our findings may provide a new approach to prevent cardiovascular aging in future spaceflights.

## 1. Introduction

As one of the major side effects, cardiovascular dysfunction has been considered as a great potential risk to astronauts’ health and performance in response to microgravity exposure [1,2,3]. Multiple mechanisms, such as arterial remodeling, hypovolemia, changes in baroreflex sensitivity, and decreases in aerobic capacity and exercise tolerance, have been demonstrated to be implicated in the occurrence of cardiovascular dysfunction after spaceflights [1,4]. Recently, aging-related changes and cardiovascular aging have been reported to accelerate in spaceflights [1]. For example, it is estimated that astronauts lose 3% of their bone density and 20% or more of their muscle mass during their three months in the International Space Station (ISS) [5,6]. By wearable sensors, oral glucose tolerance tests, blood samples, and arterial ultrasounds, one of the latest investigations indicated that carotid arteries in both male and female astronauts appeared to have aged about 20 to 30 years after six months in the ISS, as characterized by increased arterial stiffness, atherosclerosis development, and insulin resistance [7]. It is considered that cardiovascular dysfunction and aging may lead to higher blood pressure and vessel wall injury, which invoke strokes and other cardiovascular diseases [8]. Currently, exercises, particularly restive exercises, including walking and biking, are critical to build and maintain muscle and bone mass in space. Although astronauts spent 2½ h per day, six days a week exercising, the results from vessel imaging suggested that astronauts’ exercise routines were not sufficient to fully reverse the adverse effects of microgravity exposure on the vascular system, which pointed to the need for further work on appropriate countermeasures to help maintain astronauts’ health [5].

Cellular senescence is one of the causative processes of aging [9], characterized by the arrest of the cell cycle and persistent activation of p16 and p53/p21. The process of cellular senescence could be initiated by the stimulus of a series of intrinsic and extrinsic stress, including DNA damage, oxidative stress, telomere shortening, and mitochondrial dysfunction [10]. In addition, apoptosis resistance is another hallmark of cellular senescence for the significant defensive role against sublethal damage [11]. However, the apoptosis resistance-induced accumulation of intracellular waste products would ultimately activate several stresses and then aggravate the aging crisis [10,11,12]. Microgravity has a unique effect on the aging process and cellular senescence. A twin study by The National Aeronautics and Space Administration (NASA) suggested possible aging and anti-aging effects in the telomere length of leukocytes after prolonged travel aboard the ISS [13]. In addition, laboratory data indicated that simulated microgravity induced the cell G1 phase arrest and enhanced the level of oxidant stress by activating p53/p16 pathways and increasing senescence-associated beta-galactosidase expression [14]. In our previous work, we reported that activation of the large-conductance calcium-activated K^+^ channel (BK) increased apoptosis in cerebrovascular smooth muscle cells of simulated microgravity rats, which suggested BK-dependent apoptosis seemed to be a vital compensatory mechanism to alleviate the cellular senescence and aging process induced by microgravity [15].

More recently, several classes of non-coding RNAs have been discovered as key signaling regulators in the processes of cellular senescence and apoptosis resistance [16,17,18]. In addition, salidroside (SAL), a phenylpropanoid glycoside isolated from *Rhodiola rosea,* has been reported to display an important role for prevention and treatment in cardiovascular and other diseases [19] and could act as a non-coding RNA modulator to regulate cellular processes [20]. Thus, it is intriguing to explore whether SAL could target some non-coding RNAs to regulate cell apoptosis and cellular senescence and have pharmacological value in microgravity-induced aging-related changes. In the present work, we aim to investigate (1) the pharmacological value of SAL in simulated microgravity-induced cellular senescence and BK-dependent apoptosis in cerebral vascular smooth muscle cells (VSMCs) and (2) whether non-coding RNAs are targets for the therapeutic intervention of SAL in cerebral VSMCs of simulated microgravity rats.

## 2. Results

### 2.1. General Data

As indicated in Table 1, the final body weights of CON, SUS, CON + SAL, and SUS + SAL rats are not significantly different, indicating a normal growth rate during the four-week microgravity stimulation and intragastrical administration of SAL (100 mg/kg/day). Meanwhile, the ratio of soleus/body weight in both SUS and SUS + SAL rats was significantly decreased, as compared with that in CON and CON + SAL rats, respectively, suggesting the deconditioning effects of simulated microgravity in rat models.

### 2.2. SAL-Delayed Cellular Senescence in Cerebral Arteries of Simulated Microgravity Rats and Cultured VSMCs

In in vivo studies, the medial wall thickness of the basilar artery in CON, SUS, CON + SAL, and SUS + SAL rats was assessed by hematoxylin and eosin (H&E) staining (Figure 1B) and transmission electron microscopy (TEM, Figure 1C). The biomarkers of cellular senescence, including P65 and P16, were identified by Western blotting analysis (Figure 1E). As compared with that of CON rats, four-week simulated microgravity significantly increased the medial wall thickness of cerebral arteries in SUS rats (Figure 1C,D), accompanied by increased expressions of P65 and P16 (Figure 1E,F). As compared with that of SUS rats, the intragastrical administration of SAL (100 mg/kg/day) remarkably decreased the medial wall thickness of cerebral arteries in SUS + SAL rats (Figure 1C,D), accompanied by decreased expressions of P65 and P16 (Figure 1E,F). In in vitro studies, a treatment of 1000 μM H_2_O_2_ was used to induced cellular senescence by increasing mRNA levels of P65 and P16 in cultured VSMCs (Figure 1G,H)_._ After treatment with H_2_O_2_ for 1 h to induce the senescence phenotype, the following treatment of 100 μM SAL for 24 h significantly reduced the mRNA (Figure 1G,H) and protein (Figure 1I,J) expressions of P65 and P16, respectively. These results indicated that SAL treatment could attenuate the aging-like hypertrophic remodeling of cerebral arteries by delaying cellular senescence.

### 2.3. SAL-Promoted BK-Dependent Apoptosis in Cerebral Arteries of Simulated Microgravity Rats and Cultured VSMCs

Apoptosis resistance-induced accumulation of intracellular waste products would activate several stresses and aggravate the aging crisis [10,11,12]; thus, it is considered to be a hallmark of cellular senescence [11]. Here, after confirming SAL could, indeed, alleviate the H_2_O_2_-induced senescence phenotype of cultured VSMCs (Figure 1G–J), we aimed to explore the effect of SAL on VSMC apoptosis. It is reported that the large-conductance Ca^2+^ and voltage-activated K^+^ (BK) channel is a pivotal mediator for cell apoptosis [21]. In in vivo studies, as compared with those of CON rats, four-week simulated microgravity led to the protein expressions of the BK-α subunit (Figure 2A,B) and apoptotic markers, including Bax, Bcl-2, and cleaved caspase-3 (c-Caspase-3), significantly increasing in cerebral arteries of SUS rats (Figure 2C,D), accompanied by typical morphological features, such as chromatin condensation and migration to the periphery (Figure 2E). As compared with those of SUS rats, the intragastrical administration of SAL (100 mg/kg/day) further increased the protein expressions of the BK-α subunit (Figure 2A,B), Bax, Bcl-2, and cleaved caspase-3 (c-Caspase-3) in cerebral arteries of SUS + SAL rats (Figure 2C,D). In in vitro studies, treatment of SAL for 24 h induced the apoptosis of VSMCs in a dose-dependent manner, as observed by TUNEL and Hoechst staining (Figure 2F,G). The extracellular application of 100 μM SAL significantly increased the apoptosis rate, as observed by flow cytometry (Figure 2H,I), and protein expressions of the BK-α subunit, Bax, Bcl-2, and cleaved caspase-3 (c-Caspase-3) in cultured VSMCs, respectively (Figure 2J,K). In contrast, 100 nM IBTX (the specific antagonist of BK) partially inhibited the expressions of SAL-induced Bax/Bcl-2 and c-Caspase-3 (Figure 2J,K). These results suggested that SAL delayed simulated microgravity-induced cerebrovascular VSMC senescence, at least in part, by enhancing BK-dependent VSMC apoptosis and indicated that SAL treatment could promote BK-dependent apoptosis in cerebral arteries of simulated microgravity rats.

### 2.4. BK-Dependent Apoptosis Was Induced by Simulated Microgravity-Activated LncRNA-FLORPAR in VSMCs

To clarify whether non-coding RNAs participated in BK-dependent apoptosis in simulated microgravity, an Affymetrix Gene Chip Rat Transcriptome Array was used to screen differential expressions of lncRNAs between CON and SUS rats’ cerebral arteries (each group contained nine samples), which showed that 475 lncRNAs increased and 140 lncRNAs decreased in simulated microgravity (Figure 3A). But according to the prescribed statistical standard (*p*-value < 0.05 for statistical significance and fold change > 1.5 for clinical significance), no LncRNAs with significant differences were found in the sequencing results (Figure 3A), which may be attributed to the small sample size. The network plot of lncRNAs and mRNA suggested that five lncRNAs could directly target the BK-α subunit by sponging miRNAs (Appendix A and Figure 3B), in which the lncRNA-Florpar was found to increase the most obviously (Figure 3C). Next, more samples were used in the qRT-PCR analysis to confirm that the expression level of FLORPAR was significantly increased in SUS rat cerebral arteries as compared with that of CON rats (Figure 3D). The RNA fluorescence in situ hybridization (FISH) found that FLORPAR was localized in both the nuclei and cytoplasm (Figure 3E). In the gain- and loss-of-function experiments, neither overexpression of lncRNA by the pcDNA-FLORPAR vector nor inhibition of lncRNA by siRNA FLORPAR (Figure 3F) affected the mRNA expression levels of the BK-α subunit (Figure 3G). However, overexpression or inhibition of lncRNA significantly increased or decreased the protein expression of the BK-α subunit (Figure 3H,I), Bax/Bcl-2, and cleaved caspase-3 (c-Caspase-3) (Figure 3J,K) in cultured VSMCs, respectively. In addition, blocking BK with 100 nM IBTX (the specific antagonist of BK-α) markedly reduced the overexpression of lncRNA-induced apoptosis, as observed by Hoechst33324 staining, in cultured VSMCs (Figure 3L,M). In contrast, activation of BK by 30 μM NS1619 (the specific agonist of BK-α) obviously increased apoptosis rates in the presence of the inhibition of lncRNA in cultured VSMCs (Figure 3L,M). These results suggested that FLORPAR induced BK-dependent apoptosis at the post-transcriptional level.

### 2.5. MiR-193 Negatively Modulated Apoptosis by Targeting BK mRNA

Nine candidate miRNAs were screened and predicted to bind to the 3′UTR mRNA of the BK-α subunit (*KCNMA1*) by three software programs established for non-coding RNA target prediction, including miRDB (http://www.microrna.org; accessed on 15 August 2017), miRanda (http://www.microrna.org; accessed on 15 August 2017), and Targetscan(http://www.targetscan.org; accessed on 15 August 2017) (Figure 4A). By combining the microarray data of differentially expressed lncRNAs in cerebral arteries of CON and SUS rats (Figure 3A,B), we predicted that lncRNA-*FLORPAR* activated the BK-α subunit by sponging miR-193 (Figure 4B,C). Furthermore, more samples confirmed that the expression level of miR-193 significantly decreased in cerebral arteries in SUS rats as compared with those in CON rats (Figure 4D). The dual-luciferase reporter assay indicated that the miR-193 mimic significantly inhibited the relative luciferase activity of the KCNMA1 3′UTR WT reporter by co-transfecting the plasmid of KCNMA1 3′UTR NC/WT/MUT in cultured VSMCs, respectively (Figure 4E,F). In addition, the miR-193 mimic significantly reduced the protein expressions of the BK-α subunit, whereas the miR-193 inhibitor remarkably increased the protein expressions of the BK-α subunit (Figure 4G,H). Accordingly, overexpression or inhibition of miR-193 markedly decreased or increased the protein expressions of Bax/Bcl-2 and c-Caspase-3 (Figure 4I,J) in cultured VSMCs, respectively. By Hoechst33324 staining, the activation of BK by 30 μM NS1619 (specific agonist of BK-α) obviously increased apoptosis rates in the presence of the overexpression of miR-193, whereas blocking BK with 100 nM IBTX (the specific antagonist of BK-α) markedly reduced apoptosis rates in the presence of the inhibition of miR-193 in cultured VSMCs (Figure 4K,L). These results suggested that miR-193 negatively modulated the BK-α subunit by directly binding to the 3′ UTR of KCNMA1 and then inhibiting apoptosis in VSMCs.

### 2.6. MiR-193 Was Sponged by LncRNA-FLORPAR

Bioinformatic prediction suggested that the sequence of lncRNA-*FLORPAR* contained the putative binding sites of miR-193 (Figure 5A). The dual-luciferase reporter assay indicated that the miR-193 mimic distinctly reduced the relative luciferase activity of pcRNA-*FLORPAR* WT by co-transfecting the plasmid of FLORPAR NC/WT/MUT in cultured VSMCs, respectively (Figure 5B), which suggested that cytoplasmic *FLORPAR* could serve as the miR-193 sponge. Furthermore, the RNA immunoprecipitation experiment confirmed that *FLORPAR* and miR-193 were predominantly enriched in Ago2-containing miRNA ribonucleoprotein complexes, compared with those in the control IgG immune precipitate, suggesting that *FLORPAR* and miR-193 were present in the RNA-induced silencing complex (Figure 5C). Lastly, the WT *KCNMA1* 3′UTR psiCHECK^TM^-2 vector and miR-193 mimic or pcDNA-*FLORPAR* vector were co-transfected, respectively, which showed that the miR-193 mimic distinctly suppressed the WT *KCNMA1* 3′UTR activity, whereas *FLORPAR* could partially reverse the miR-193-inhibited activity of WT KCNMA1 3′UTR (Figure 5D) in cultured VSMCs. These results clearly indicated that lncRNA FLORPAR could have biological functions by sponging miR-193.

### 2.7. SAL Activated LncRNA-FLORPAR/miR-193 Pathway and Then Induced BK-Dependent Apoptosis in VSMCs

As shown in Figure 6A,B, the inhibition of LncRNA-FLORPAR by Si-FLORPAR, activation of miR-193 by the miR-193 mimic, or inhibition of the BK-α subunit by IBTX significantly decreased SAL-induced apoptosis in cultured VSMCs, which suggested that SAL increased BK-dependent apoptosis by modulating LncRNA-FLORPAR and miR-193.

## 3. Discussion

The major and novel findings of this work were: (1) salidroside delayed cellular senescence and increased the BK-dependent apoptosis of VSMCs in vivo and in vitro; (2) simulated microgravity stimulated LncRNA-FLORPAR-induced BK-dependent apoptosis by sponging miR193 in VSMCs; (3) SAL increased BK-dependent apoptosis by activating LncRNA-FLORPAR-sponging miR-193 pathway, which suggested that SAL is a novel non-coding RNA modulator and may provide a new approach to prevent cardiovascular aging in future spaceflights.

### 3.1. SAL Alleviated Simulated Microgravity-Induced Arterial Hypertrophic Remodeling by Delaying Cellular Senescence and Promoting BK-Dependent Apoptosis in VSMCs

Generally, arterial pressure derives from three kinds of energy: elastic (owing to the blood volume and vascular structure), kinetic (owing to the velocity of the flowing blood), and gravitational. The gravitational component generates hydrostatic pressure gradients throughout circulation [22], which would be eliminated by microgravity and cause an increased transmural pressure in the cerebral circulation [1]. During microgravity exposure, cerebrovascular arteries would launch the enhancement of the myogenic tone and vasoconstrictor reactivity [4] to maintain the downstream microcirculation [23,24,25]. Recently, several human studies based on ISS onboard missions or spaceflights have revealed that microgravity accelerated the aging process and cellular senescence in the cardiovascular system, which may contribute to the structural and functional deterioration of vessel walls [7,13,14]. Senescent cells are highly resistant to apoptosis and exhibit a high level of proliferative potential by releasing pro-inflammatory molecules. Apoptosis is regarded as a kind of cell clearance mechanism to eliminate damaged and dysfunctional cells [26,27], in which the activation of BK may trigger the initial step of apoptosis by decreasing the concentration of intracellular K^+^ and resulting in cell shrinkage (apoptotic volume decrease) in VSMCs [28]. The present work found that simulated microgravity induced an obvious hypertrophic remodeling (Figure 1B–D) with increased expressions of p53 and p16, the senescence markers (Figure 1E,F, respectively), in rat cerebral arteries. But, surprisingly, BK-dependent apoptosis was significantly increased in the cerebral arteries of simulated microgravity rats (Figure 2A–E), which is considered as a compensatory mechanism, during microgravity exposure, against cerebrovascular hypertrophic remodeling. It is reported that the responses of some major physiological systems, during microgravity exposure, showed significant differences in each system over time and associated risks [4]. Some are adaptive changes, such as region-specific vascular remodeling, which promotes physiological function in a weightless environment and usually stabilizes after a period of about 4–6 weeks. Vascular adaptations to microgravity suggest some forceful compensatory mechanisms of the cardiovascular system could be mobilized in microgravity to prevent the irreversible and progressive injury of vessels [25,26,27].

SAL is a bioactive tyrosine-derived phenolic natural product found in Rhodiola genus plants [19]. It has been reported that SAL exerts a significant pro-apoptosis effect in multiple disease models, including hyperglycemia [29], gastric cancer [30], hepatocellular carcinoma [31], spinal cord injury [32], ischemic stroke [33], and pulmonary arterial hypertension [34]. In in vivo and in vitro studies, we found that the administration of SAL remarkably alleviated the cerebrovascular hypertrophic remodeling of simulated microgravity rats (Figure 1B–D) by decreasing the expressions of senescence markers (p53 and p16, Figure 1E–J) and increasing the BK-dependent apoptosis of VSMCs (Figure 2A–K). These results suggested that SAL delayed cellular senescence and potentiated the compensatory and protective effects of BK-dependent apoptosis in VSMCs against cerebrovascular remodeling during exposure to microgravity. There was inconsistency in SAL inducing apoptosis in in vivo and in vitro models, in which BK-dependent apoptosis was unmodified upon the administration of SAL in control rats; but in the in vitro smooth muscle cell line without any treatment, BK-dependent apoptosis was activated by SAL administration. But considering the subjects, routes, and action concentrations of administered SAL were all different in the in vivo and in vitro experiments, it was hard to guarantee that the results would be completely consistent. In addition, under the condition of the in vitro culture, cells usually have a certain basic apoptosis rate. Just as reported in our present (Figure 2I) and previous [28] studies, even without SAL or any other treatment, A7r5 cells cultured in vitro also had a basic apoptosis rate of about 5–7%. Thus, similar to the in vivo results, for which a microgravity stimulus was needed to induce BK-dependent apoptosis, the in vitro results indicated that SAL potentiated basic (in vitro culture condition-induced) BK-dependent apoptosis.

### 3.2. LncRNA-FLOPPAR-Sponging miR-193 Pathway Induced BK-Dependent Apoptosis in VSMCs of Simulated Microgravity Rats

Long non-coding RNAs (>200 nt) have been demonstrated to repress or activate gene transcription by modulating chromatin activity in nuclei or regulating mRNA and protein stability in the cytoplasm [35]. Short non-coding RNAs (<200 nt), such as miRNA, could bind with their target mRNA and then induce mRNA degradation or repress translation [36]. For example, LncRNA-H19 could promote VSMC proliferation and suppress apoptosis by sponging miR-148b and then modulating the WNT/β-catenin pathway [37]. In addition, it has been reported that LncRNA-GAS5-sponging miR-21 could block Akt phosphorylation and LncRNA-Tug1-sponging miR-374c could stimulate Notch signaling [38,39]. In the present work, gene chips and sequence-alignment predictions were used for screening, and we found that lncRNA-FLORPAR (Figure 3A–E) promoted the apoptotic signaling pathway (Figure 3J–M) by activating the BK-α subunit at the protein level, not at the mRNA level (Figure 3F–I) in the cerebral arteries of simulated microgravity rats. Furthermore, bioinformatic predictions and the dual-luciferase reporter assay suggested that miR-193 may be sponged by LncRNA-FLORPAR (Figure 4A–D) and then bind the site of KCNMA1 (the mRNA of the BK-α subunit) 3′UTR (Figure 4E,F) to regulate BK-dependent apoptosis in VSMCs (Figure 4G–L). In addition, the RNA immunoprecipitation experiment indicated that LncRNA-FLORPAR could directly sponge the site of miR-193 to form RISC (Figure 5A–C) for regulating apoptosis in VSMCs (Figure 5D). Therefore, we found a new pathway of non-coding RNAs, in which LncRNA-FLORPAR could promote BK-dependent apoptosis by sponging miR-193 in VSMCs exposed to simulated microgravity.

### 3.3. SAL Promoted BK-Dependent Apoptosis by Activation of LncRNA-FLORPAR/miR-193 Pathway

Accumulating evidence indicated that SAL displayed a wide range of pharmacological activities through the regulation of non-coding RNAs. For example, SAL has been reported to reduce the activity of miR-323-3p and then promote the transcription of the suppressor of cytokine signaling 5 (SOCS5), thereby attenuating airway inflammation and remodeling in asthmatic mice [40]. In addition, SAL has been demonstrated to prevent hypoxia-induced damage in human retinal endothelial cells via the miR-138/ROBO4 Axis [20]. In the present work, we found that SAL is a novel non-coding RNA modulator, which promoted apoptosis in VSMCs by activating LncRNA-FLORPAR, inhibiting miR-193, and then triggering the activity of the BK-α subunit (Figure 6). Taken together, simulated microgravity induced age-like hypertrophic remodeling in rat cerebral arteries. The administration of SAL plays a protective role to delay cellular senescence and promote BK-dependent apoptosis by modulating the LncRNA-FLORPAR/miR-193 pathway in VSMCs. 

### 3.4. Practical Implications and Limitations

Currently, exercise is the main preventive measure against accelerated aging processes in space; however, it is not sufficient to fully reverse the adverse effects of microgravity [5]. Our present study found, for the first time, that SAL treatment could delay vascular aging under the simulated microgravity condition, suggesting a potential pharmacological value for SAL to reduce cardiovascular risk in astronauts. In addition, considering the unique effects of microgravity on the cardiovascular system, spaceflight studies provide a new platform to explore the potential mechanisms of cardiovascular aging and to develop effective interventions. However, there are still some limitations in our current work. The present work was conducted on rat models of simulated microgravity but lacked evidence of human bodies in a real microgravity environment. And, although the tail-suspension hindlimb-unloaded rat model is the most generally accepted rodent model for simulated microgravity experiments, cautions should be taken in the following aspects: (1) The entire body is unloaded during spaceflights, whereas the forelimbs, head, and upper back remain weightbearing in hindlimb-unloaded animals; (2) hindlimb unloading does not completely eliminate hydrostatic gradients but only changes their directions and magnitudes; (3) fluid shifts may be stronger in hindlimb-unloaded rats because the animal is a quadruped, and fluid redistribution may be limited in rats during spaceflights [41,42]. In addition, the mechanism through which non-coding RNAs sense microgravity stress needs further investigations.

## 4. Materials and Methods

### 4.1. Animals

All the animal experiments were conducted according to the guidelines of the Institutional Animal Care and Use Committee of the Air Force Military Medical University Approval code: 20220344). Rats (six rats in each group) were randomly divided into four groups: (1) the control group (CON), (2) control rats treated with salidroside (SAL), (3) tail-suspension rats (SUS), and (4) tail-suspension rats treated with salidroside (SUS + SAL). The modified suspension techniques [43,44], with which Sprague–Dawley rats (weight: 210–230 g) were maintained at about a −30° head-down tilt position with their hindlimbs unloaded for 28 days, were used to simulate the cardiovascular effects of microgravity in the SUS and SUS + SAL groups. The left soleus/body weight radio was measured to confirm the simulated microgravity efficiency in tail-suspended (SUS) rats routinely. Rats in the SAL and SUS + SAL groups were continuously administered salidroside (SAL) (100 mg/kg/day, dissolved in distilled water) by the intragastric (i.g.) route for 28 days.

### 4.2. Isolation of Cerebral Arteries 

The rats were anesthetized with pentobarbital sodium (50 mg/kg, i.p.) and killed by exsanguination via the abdominal aorta. Briefly, the brain tissue was carefully isolated, first, and placed in a 4 °C phosphate-buffered saline solution (PBS). Then, under a stereomicroscope, the meninges were carefully removed, and the cerebral arteries, including the basilar arteries; anterior, middle, and posterior cerebral arteries; and the Willis circle arteries, were collected.

### 4.3. Cell Cultures

The A7r5 cell line is a kind of spontaneously immortalized cell line derived from the thoracic aorta of embryonic rats, which could adapt to continuous culture conditions and possesses the properties of smooth muscle cells. Thus, it is a classical smooth muscle cell model and has been widely used to study cellular and molecular signaling mechanisms and the effects of various drugs and hormones on VSMCs [45]. In the present study, A7r5 cells were purchased from the Chinese Academy of Sciences (Shanghai, China) and cultured in DMEM (Hyclone, Logan, UT, USA) supplemented with 10% fetal bovine serum (FBS) (Thermo Scientific, Rockford, IL, USA), 100 U/mL penicillin (Solarbio, Beijing, China), and 100 µg/mL streptomycin (Solarbio, Beijing, China). The cells were cultured at 37 °C in an atmosphere of 5% CO_2_ and routinely sub-cultured every 48 h.

### 4.4. Protein Extraction and Western Blotting

The cerebral arteries and A7r5 cell lysis buffer were prepared in M-PER Mammalian Protein Extraction Reagent (Thermo Scientific, Rockford, IL, USA) with a fresh 1% protease inhibitor cocktail (Thermo Scientific, Rockford, IL, USA). After centrifugation, the supernatants were denatured for Western blotting. Proteins were separated using NuPAGE 4–12% Bis-Tris gel (Thermo Scientific, Rockford, IL, USA) and then transferred to polyvinylidene fluoride (PVDF) membranes (Millipore, Billerica, MA, USA). The membranes were blocked and subsequently incubated with appropriate primary antibodies against Caspase-3 (1:1000; CST#9664; Cell Signaling Technology, Danvers, MA, USA), Bax (1:1000; CST#41162S; Cell Signaling Technology, Danvers, MA, USA), Bcl-2 (1:1000; ab117115; Abcam, Eugene, OR, USA), GAPDH (1:50,000; Cat No. 60004-1-Ig; Proteintech, Wuhan, China), BKCa α-subunit (1:500; APC-107; Alomone Labs, Jerusalem, Israel), P16 (1:2000; Cat No.10883-1-AP; Proteintech, Wuhan, China), and P53 (1:5000; Cat No. 60283-2-Ig; Proteintech, Wuhan, China) at 4 °C overnight. The membranes were then incubated for 2 h with horseradish peroxidase (HRP)-conjugated secondary antibodies (1:10,000; Cat No. SA00001-2 and Cat No. SA00001-1; Proteintech, Wuhan, China), and proteins were detected and visualized using the chemiluminescent HRP substrate (Millipore, Billerica, MA, USA). Image J software version 1.8.0 was applied for quantification.

### 4.5. RNA Sequencing

RNA sequencing and library construction were performed by the Shanghai Qiming Information Technology Company. In brief, sequence libraries were prepared using the NEB Next Ultra Directional RNA Library Prep Kit for Illumina (NEB, Ipswich, MA, USA) selection to include all the lncRNA transcripts that were not polyadenylated. Libraries were sequenced on the microarray Gene Chip (Affymetrix Gene Chip Rat Transcriptome Array 1.0) according to the manufacturer’s protocol.

### 4.6. Analysis of Non-Coding RNA Target

MiRNAs that were targeted by lncRNA-FLORPA and could bind to *KCNMA1* (the mRNA of the BKCa α-subunit) were predicted using the TargetScan database (version 6.0) (http://www.targetscan.org/; accessed on 15 August 2017), miRanda (http://www.microrna.org/microrna/home.do; accessed on 15 August 2017), and miRDB (http://www.mirdb.org/; accessed on 15 August 2017).

### 4.7. RNA Extraction and Real-Time Quantitative Reverse Transcription PCR (qRT-PCR)

Cerebral artery samples and A7r5 cells were mixed with RNAiso (Takara, Otsu, Japan) and homogenized by grinding. After centrifugation, phase separation, and precipitation, the resulting RNA pellet was dissolved in RNase-free water. For the qRT-PCR assay, the total RNA, including lncRNAs and miRNAs, was reverse-transcribed to cDNA using a Mir-X miRNA First-Strand Synthesis Kit (Takara, Otsu, Japan) according to the manufacturer’s protocol. Then, the cDNA was amplified with SYBR Premix Ex TaqTM (Takara, Otsu, Japan) using a CFX96 (Bio-rad, Richmond, CA, USA) instrument. Data were analyzed via the relative Ct (2^−ΔΔCt^) method and expressed as a fold change compared with the respective control. The pairs of primers used in amplification included: LncRNA-Florpar (forward-CCGTGGCTTCTAGTGGGATA and reverse-GGAACTGCAAATCCTGTGCT), BKca α-subunit (forward-GGAGGATGCCTCGAATATCA and reverse-AGCTCGGGATGTTTAGCAGA), BKca β-subunit (forward-TCTGTTGCAGGACTAACCTT and reverse-GAGCTGCTGTTGCTCTTATT), P16 (forward-GCCTTTTCACTGTGTTGGAG and reverse-TGCCATTTGCTAGCAGTGTG), P53 (forward-TGCTGAGTATCTGGACGACA and reverse-CAGGCACAAACACGAACC), miR-193 (forward-AACTGGCCTACAAAGTC and reverse-GTGCAGGGTCCGAGGT), β-actin (forward-AAAGAAAGGGTGTAAAACGCA and reverse-TCAGGTCATCACTATCGGCAAT), and U6 (forward-CTCGCTTCGGCAGCACA and reverse-AACGCTTCACGAATTTGCGT).

### 4.8. Plasmid and Oligonucleotide Transient Transfection

Overexpression plasmid pcDNA-lncR-FLORPAR, siRNA-targeting lncR-FLORPAR, miR-193 mimic/inhibitor, and corresponding controls were designed and synthesized by Ribobio Corporation (Guangzhou, China). A7r5 cells were transfected with the DNA plasmid and oligonucleotides using the Lipofectamine 3000 reagent (Invitrogen, Carlsbad, CA, USA) and Opti-MEM Reduced-Serum medium (Invitrogen) and harvested after 48 h of transfection according to the manufacturer’s instructions. The sequences were as follows: lncR-FLORPAR siRNA-1, GCAGTTATCACAGGAGTTT; lncR-FLORPAR siRNA-2, CAAGAT CTGAGAGGAGAA; lncR-FLORPAR siRNA-3, CAACTTCCAAATCTAAAGA; lncR-FLORPAR siRNA-4, CAGACCACGCACACTCTCCT; miR-195-3p mimic, 5′-CCAAUAUUGGCUGUGCUGCUGCUCC-3′ (sense) and 5′-AGCAGCACAGCCAAUAUUGGUU-3′(antisense); and miR-195-3p inhibitor, 5′-GGAGCAGCACAGCCAAUAUUGG-3′.

### 4.9. Dual-Luciferase Report Assay

The PsiCHECK^TM^-2 vector (Promega, Madison, WI, USA), which contains both Firefly and Renilla luciferase genes, was used to introduce the wild/mutant 3′ UTR sequences of KCNMA1 and wild/mutant sequences of lncR-FLORPAR immediately downstream of the stop codon of the Renilla luciferase gene downstream to create a wild-type (WT) or mutant-type (MUT) KCNMA1 3′ UTR plasmid (Sangon Biotech, Shanghai, China) and a WT or MUT lncR-FLORPAR plasmid (Sangon Biotech, Shanghai, China), respectively. The PsiCHECK™-2 vector without the inserted gene was used as the negative control (NC) plasmid. According to the manufacturer’s protocols, A7r5 cells were co-transfected with the PsiCHECK^TM^-2 vector (WT/MUT/NC) and oligonucleotides (miR-195 mimic or mimic control) using the Lipofectamine 3000 reagent. The cells were lysed after 48 h of transfection, and both the Firefly and Renilla luciferase activities (Fluc, Rluc) were sequentially measured using the Dual-Luciferase Reporter Assay system (Promega). The relative luciferase activity was calculated by normalizing Rluc to Fluc, and the value in the NC plasmid plus the mimic control-treated group was set at one.

### 4.10. RNA Fluorescence In Situ Hybridization (FISH)

RNA fluorescence in situ hybridization was performed using a fluorescence in situ hybridization kit (Ribobio Corporation, Guangzhou, China), following the manufacturer’s instructions, to determine the localization of lncRNA-FLORPAR in A7r5 cells. All the probes were labeled with CY3 fluorescent dye, and fluorescence detection was performed with a Zeiss LSM 800 confocal microscope.

### 4.11. RNA Immunoprecipitation (RIP)

An RIP experiment was performed using an EZ Magna RIP kit (Millipore, Billerica, MA, USA) following the manufacturer’s protocol. A7r5 cells were lysed in complete RIP lysis buffer. The extract was incubated with the Ago2 antibody or control IgG (Millipore, Billerica, MA, USA) for 6 h at 4 °C, followed by adding magnetic beads conjugated with protein A/G (Thermo Fisher Scientific, Waltham, MA, USA). The beads were washed and incubated with Proteinase K to remove the proteins. Finally, the purified RNA was subjected to qRT-PCR analysis.

### 4.12. Hoechst Staining

Hoechst 33342 was used to observe nuclear morphological aspects by fluorescence microscopy using a filter from 320 to 350 nm. A7r5 cells were washed twice in phosphate-buffered saline, fixed in phosphate-buffered saline containing 1% (wt/vol) paraformaldehyde (Fisher Scientific, Pittsburgh, PA, USA), rinsed with water, and stained with Hoechst 33342 (final concentration, 5 μg/mL) for 25 min. After staining with Hoechst 33342, the morphological aspects of the nuclei were observed with fluorescence microscopy (Leica DMi8-M, Leica AG, Wetzlar, Germany).

### 4.13. Terminal Deoxynucleotidyl Transferase dUTP Nick End-Labeling (TUNEL) Assay

Apoptotic cells were detected using a TUNEL apoptosis detection kit (Sigma, Concord, CA, USA) according to the manufacturer’s instructions. A7r5 cells were fixed by 4% PFA in PBS for 25 min, incubated in an equilibration buffer for 10 min and then with PBS containing Triton X-100 for 5 min, followed by the TUNEL reaction system at 37 °C for 1 h in the dark. Finally, the cells were stained with Hoechst 33342 for 5 min, and an in situ cell death detection kit was used to detect apoptosis. Images were captured using a fluorescence microscope (Leica DMi8-M).

### 4.14. Flow Cytometry

Apoptotic cells were detected using the Annexin V-PE/FITC Apoptosis Detection Kit (Invitrogen, Carlsbad, CA, USA), according to the manufacturer’s instructions, and analyzed using FACSCanto II Flow cytometry (BD, Franklin Lakes, NJ, USA). The data were analyzed using CELLQuest software version 3.0.

### 4.15. Transmission Electron Microscopy (TEM) 

Basilar arteries were fixed, dehydrated, and embedded in epoxy resin. Ultrathin sections were obtained with a diamond knife using an ultramicrotome (EMUC6/FC6; Leica Microsystems, Wetzlar, Germany) and stained with lead citrate and uranyl acetate. Next, the sections were viewed and photographically recorded using a scanning TEM (Tecnai G2 F20S-TWIN; FEI, Columbia, MD, USA) operating at 100 kV with a high-performance Gatan CCD camera (AMETEX, Berwyn, PA, USA). For morphological assessments, individual cerebral VSMCs, which reside in the tunica media of basilar arteries, were observed after being magnified 6000 times.

### 4.16. Hematoxylin-Eosin (H&E) Staining

Cerebral arteries were fixed in 4% paraformaldehyde at room temperature and then embedded in paraffin. After routine deparaffinization and rehydration procedures, sections, 4–5 μm thick, were stained with hematoxylin (PL-0125-HEM, PatoLab, Istanbul, Turkey) and then analyzed under a light microscope (Olympus, Tokyo, Japan).

### 4.17. Statistical Analysis

All the data are presented as means ± SEM. Data processing and statistical analyses were performed using GraphPad Prism software (version 8.0, GraphPad Software Inc., San Diego, CA, USA) and either an unpaired *t*-test (two group comparison) or one-way ANOVA using Tukey’s post hoc test (multiple group comparison). For relative gene expressions, the mean value for the control group was defined as one. A *p*-value < 0.05 was considered as statistically significant.

## 5. Conclusions

In summary, our present work suggests that SAL is a novel non-coding RNA modulator for regulating the LncRNA-FLORPAR-sponging miR-193 pathway, which significantly promotes BK-dependent apoptosis and delays cerebrovascular aging-like remodeling in VSMCs during simulated microgravity exposure. Our findings may provide a new approach to prevent cardiovascular aging in future spaceflights.

## Figures and Tables

**Figure 1 ijms-24-14531-f001:**
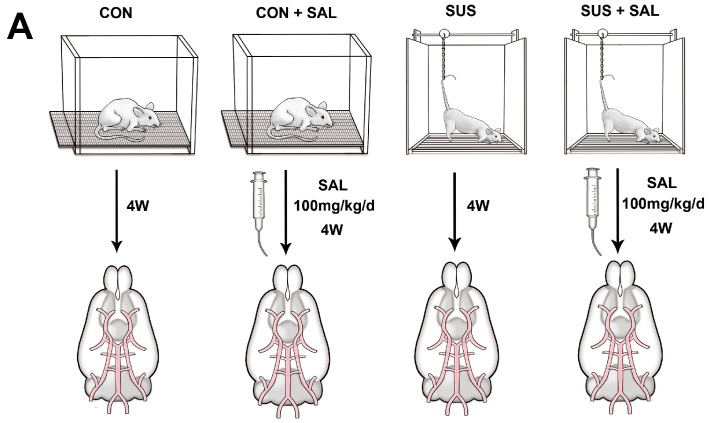
Salidroside-delayed cellular senescence in VSMCs in vivo and in vitro. (**A**) Schematic diagram of animal experiments. Rats were randomly divided into four groups: (1) control group (CON), (2) control rats treated with salidroside (100 mg/kg/day) (SAL), (3) tail-suspension rats (SUS), and (4) tail-suspension rats treated with salidroside (100 mg/kg/day) (SUS + SAL). After four weeks of treatment, the cerebral arteries of the rats were taken for further analysis. (**B**,**C**): Representative photomicrographs of basilar arterial remodeling indicated by H&E staining (**B**) and transmission electron microscopy (TEM) observation (**C**) after 100 mg/kg/day salidroside (SAL) treatment in four-week simulated microgravity rats. (**D**) Morphometric analysis of the medial wall thickness of the basilar artery in CON, SUS, CON + SAL, and SUS + SAL rats. (**E**,**F**) Western blotting indicated the senescence-related protein expression, including P65 and P16, in cerebral arteries after 100 mg/kg/day salidroside (SAL) treatment in four-week simulated microgravity rats. (**G**,**H**) Effect of salidroside (SAL) from 25 μM to 500 μM on cellular senescence induced by 1 mM H_2_O_2_ in cultured VSMCs. The mRNA expressions of senescence-related genes: P53 (**G**) and P16 (**H**) were examined using qRT-PCR. (**I**,**J**): Western blotting indicated the protein expressions of P53 and P16 in SAL (100 μM)-treated senescent A7r5 cells. Data are presented as means ± SEM. n = 6/group, * *p* < 0.05 vs. CON, # *p* < 0.05 vs. SUS/H_2_O_2,_ ** *p* < 0.01 analyzed by one-way ANOVA.

**Figure 2 ijms-24-14531-f002:**
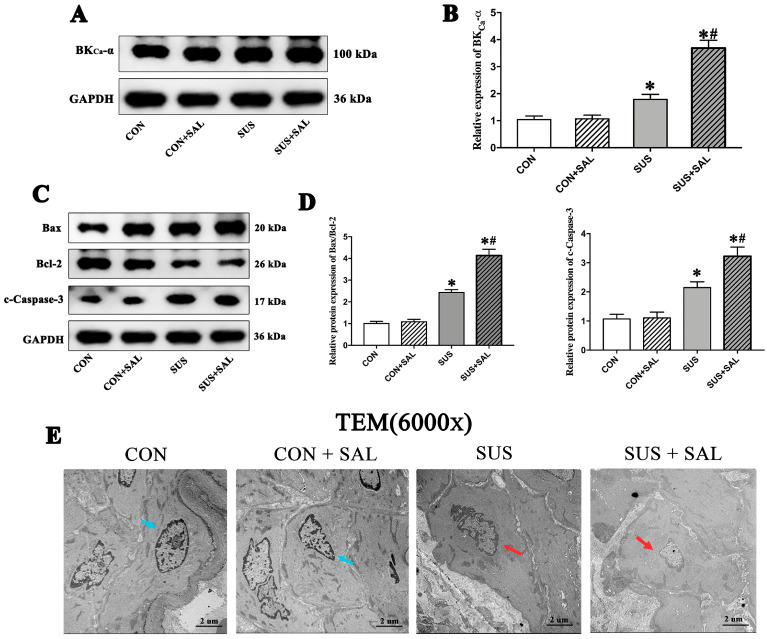
Salidroside increased BK-dependent apoptosis in VSMCs in vivo and in vitro. (**A**,**B**) Expression of BK-α subunit in rats’ cerebral arteries after 100 mg/kg/day salidroside (SAL) treatment in four-week simulated microgravity rats. (**C**,**D**) Western blotting indicated the apoptosis-related protein expressions, including Bax, Bcl-2, and cleaved caspase-3 (c-Caspase-3), in cerebral arteries after 100 mg/kg/day salidroside (SAL) treatment in four-week simulated microgravity rats. (**E**) Comparison of VSMC apoptosis in cerebral arteries of CON, SUS, CON + SAL, and SUS + SAL rats observed using transmission electron microscopy. Blue arrows show the normal nucleus; red arrows show the apoptotic nucleus. (**F**) Effects of salidroside (SAL) from 25 μM to 500 μM on VSMC apoptosis investigated by TUNEL (**upper**) and Hoechst (**lower**) staining. White arrows show the apoptotic cells. (**G**) Apoptosis rate was in a dose-dependent manner after SAL treatment. (**H**,**I**) Apoptotic cell percentages after SAL treatment was examined by flow cytometry. (**J**,**K**) Western blotting indicated the expressions of BK-α subunit, Bax, Bcl-2, and c-Caspase-3 in the treatment of SAL, respectively. Data are presented as means ± SEM. ** *p* < 0.01 vs. CON, * *p* < 0.05 vs. CON, # *p* < 0.05 vs. SUS, ^&^ *p* < 0.05 vs. 0 μM, analyzed using Student’s *t* test in (**I**) and analyzed by one-way ANOVA in (**B**,**D**,**G**,**K**). Scale bars: 2 μm.

**Figure 3 ijms-24-14531-f003:**
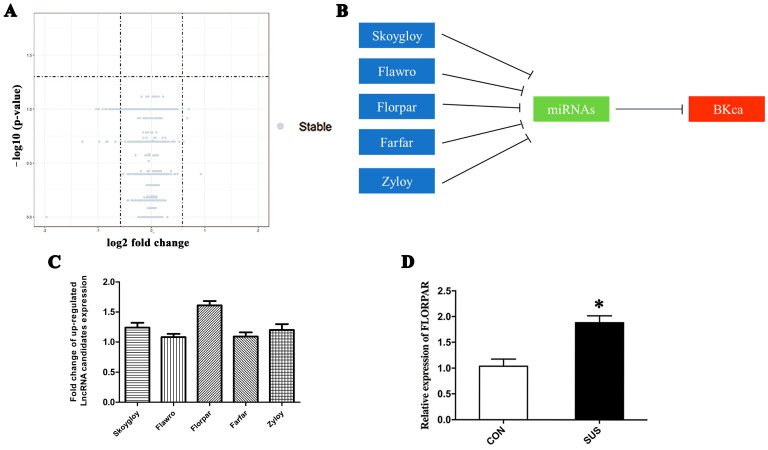
LncRNA-FLORPAR induced apoptosis by activation of BK-α subunit at protein level, not at mRNA level. (**A**) Volcano map analysis shows the differentially expressed LncRNAs in cerebral arteries of CON and SUS rats. Differentially expressed LncRNAs were compared by R package limma and Benjamini. False Discovery Rate (FDR) was used to adjust *p* values with *p*-value < 0.05 for considering statistical and clinical significance defined as a difference of 1.5 folds. (**B**) Model diagram of predicted LncRNAs, which could sponge miRNAs and then target BK-α mRNA. (**C**) Expression levels of candidate LncRNAs in cerebral arteries of simulated microgravity rats screened by transcriptome microarray gene chip. (**D**) qRT-PCR was used to indicate expressions of FLORPAR in CON and SUS rat cerebral arteries, n = 10/group. (**E**) RNA fluorescence in situ hybridization (FISH) for FLORPAR in cultured VSMCs. Nuclei, blue; FLORPAR, red. U6 was detected as the positive control for nucleus locations, whereas 18 S was detected as the positive control for cytoplasm location. Scale bars: 10 μm. (**F**) qRT-PCR indicated expressions of LncRNA-FLORPAR after 48 h of transfection with pcDNA empty vector (pcDNA-NC), pcDNA-FLORPAR, and siRNA negative control (Si-NC) or Si-FLORPAR, respectively. (**G**) qRT-PCR showed the mRNA expression of the BK-α subunit after transfection with pcDNA-FLORPAR or Si-FLORPAR. (**H**,**I**) Western blotting indicated the protein level of the BK-α subunit after pcDNA-FLORPAR or Si-FLORPAR transfection in cultured VSMCs, respectively. (**J**,**K**) Western blotting indicated the protein levels of Bax, Bcl-2, and cleaved caspase-3 (c-Caspase-3) after pcDNA-FLORPAR or Si-FLORPAR transfection, respectively. (**L**,**M**) Treatments of 100 nM IBTX or 30 μM NS-1619 were performed for 24 h in the transfection of pcDNA-FLORPAR or Si-FLORPAR. Apoptotic cells were observed and quantitated by Hoechst staining. Scale bars: 20 μm. Data are presented as means ± SEM. n = 3/group, * *p* < 0.05 vs. CON/pcDNA-NC, # *p* < 0.05 vs. Si-NC, & *p* < 0.05 vs. pcDNA-FLORPAR, $ *p* < 0.05 vs. Si-FLORPAR analyzed using Student’s *t* test in (**D**) and analyzed by ANOVA in (**C**,**F**,**G**,**I**,**K**,**M**).

**Figure 4 ijms-24-14531-f004:**
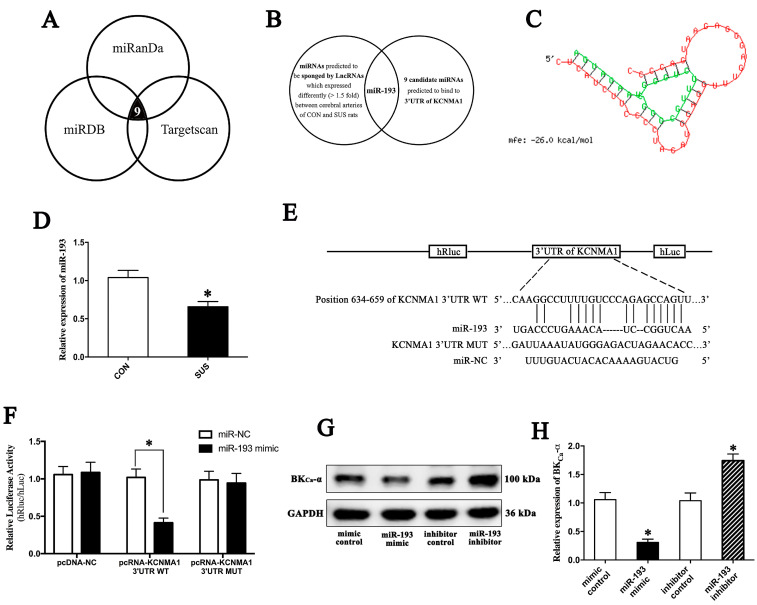
MiR-193 inhibited apoptosis by targeting BK mRNA. (**A**) Prediction of 9 candidate miRNAs, which may target BK-α subunit mRNA. (**B**) Prediction of candidate miRNAs, which could be sponged by differentially expressed LncRNAs in SUS and CON rats. (**C**) Bioinformatics predicted the binding site of LncRNA-Florpar with miR-193. (**D**) qRT-PCR indicated the mRNA expression of miR-193 in cerebral arteries of CON and SUS rats, n = 10/group. (**E**) Schematic diagram of the binding sequences of miR-193 to KCNMA1 3′UTR. (**F**) Dual-luciferase activity assay was carried out by co-transfecting miR-193 mimic and psiCHECKTM-2 vectors, which contained WT/MUT KCNMA1 3′UTR sequences, in cultured VSMCs. The relative luciferase activity refers to the value of hRluc/hLuc (n = 3/group). (**G**,**H**) Western blotting showed the protein expression of the BK-α subunit in cultured VSMCs transfected with miR-193 mimic or inhibitor (n = 3/group). (**I**,**J**) Western blotting indicated the expressions of Bax, Bcl-2, and cleaved caspase-3 (c-Caspase-3) in cultured VSMCs transfected with miR-193 mimic or inhibitor (n = 3/group). (**K**,**L**) Treatment of 30 μM NS-1619 or 100 nM IBTX was performed for 24 h after miR-193 mimic or inhibitor transfection for 48 h. Apoptotic cells were observed and quantitated by Hoechst staining. Data are presented as means ± SEM (n = 3/group). * *p* < 0.05 vs. mimic/inhibitor control, # *p* < 0.05 vs. miR-193 mimic/inhibitor analyzed using Student’s *t* test in (**D**,**F**,**H**,**J**) and analyzed by ANOVA in (**L**). Scale bars: 20 μm.

**Figure 5 ijms-24-14531-f005:**
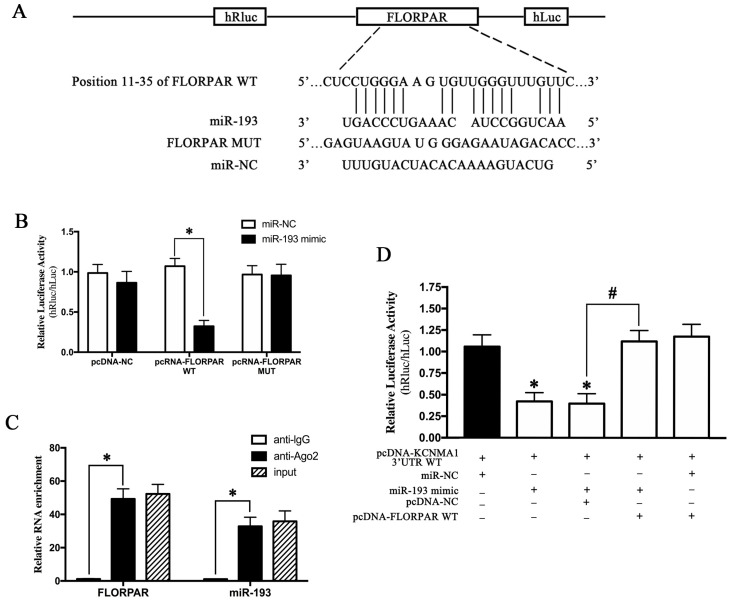
MiR-193 was directly sponged by LncRNA-FLORPAR. (**A**) Schematic diagram of the binding sequences of miR-193 to FLORPAR. (**B**) Dual-luciferase activity assay was carried out by co-transfecting miR-193 mimic and psiCHECKTM-2 vectors, which contain the WT/MUT FLORPAR sequence in A7r5 cells. The relative luciferase activity refers to the value of hRluc/hLuc (n = 3/group). (**C**) Lysates of cultured VSMCs were used for RIP with an Ago2 antibody and an IgG antibody. The qRT-PCR showed the levels of LncRNA-FLORPAR and miR-193 (n = 3/group). (**D**) psiCHECKTM-2 vector, containing WT KCNMA1 3′UTR sequences, and miR-193 mimic/control were co-transfected with pcDNA-FLORPAR or the NC vector to confirm the competing endogenous RNA activity of FLORPAR. The ratio of hRluc/hLuc in the WT KCNMA1 3′UTR psiCHECKTM-2 vector after 48 h of transfection was applied as the relative luciferase activity (n = 3/group). Data are presented as means ± SEM. * *p* < 0.05 vs. relative control, # *p* < 0.05 vs. pcDNA-NC analyzed using Student’s *t* test in (**B**) and analyzed by ANOVA in (**C**,**D**).

**Figure 6 ijms-24-14531-f006:**
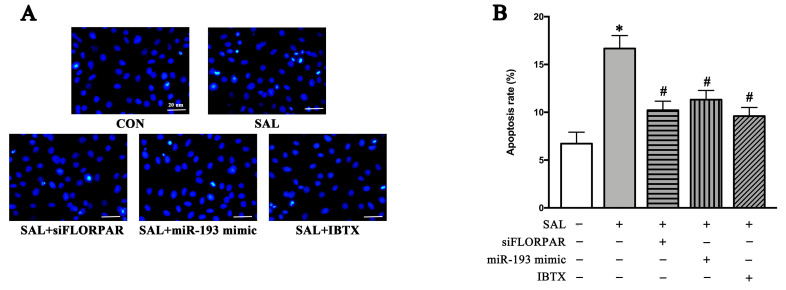
SAL increased BK-dependent apoptosis by regulating FLORPAR and miR-193. (**A**,**B**) Apoptotic A7r5 cells were detected by Hoechst33324 staining after 24 h of SAL treatment in the presence of Si-FLORPAR, miR-193 mimic transfection, and IBTX (the specific antagonist of the BK-α subunit) (**A**) and quantified by apoptosis rate (**B**). Data are presented as means ± SEM. n = 3/group, * *p* < 0.05 vs. CON, # *p* < 0.05 vs. SAL analyzed by one-way ANOVA. Scale bars: 20 μm.

**Table 1 ijms-24-14531-t001:** Body weights, soleus wet weights, and ratios of soleus/body weight in CON, SUS, CON + SAL, and SUS + SAL rats.

	Body Weight (g)		Left Soleus Weight (mg)	Soleus/Body Weight (mg/g)
	Initial	Final		
CON	216.65 ± 4.26	405.38 ± 7.42	151.29 ± 4.04	0.38 ± 0.02
SUS	220.28 ± 5.34	399.21 ± 6.93	70.16 ± 2.96 **	0.18 ± 0.01 **
CON + SAL	218.45 ± 4.68	404.36 ± 7.74	147.43 ± 3.35	0.39 ± 0.02
SUS + SAL	222.42 ± 4.15	402.32 ± 7.17	72.42 ± 3.29 **	0.18 ± 0.01 **

Values of body weights are means ± SD; others are means ± SEM. CON: 28-day control rats; SUS: 28-day tail-suspended rats; CON + SAL: CON rats administered salidroside; SUS + SAL: SUS rats administered salidroside; n = 32, ** *p* < 0.01 vs. CON (analyzed by one-way ANOVA).

## Data Availability

All the data generated by the study are included in the manuscript or Appendix A.

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
