# Peer review of "Discovery of Salidroside as a Novel Non-Coding RNA Modulator to Delay Cellular Senescence and Promote BK-Dependent Apoptosis in Cerebrovascular Smooth Muscle Cells of Simulated Microgravity Rats"

_ijms, 2023, doi:10.3390/ijms241914531_

Round 1

Reviewer 1 Report

The manuscript submitted by Yi-Ling Ge et al. is presented as the study of the mechanism by which salidroside (SAL) delays vascular smooth muscle cell senescence of simulated microgravity rats. The most valuable contribution of this work is the finding of a BK channel post-transcriptional modulation by pair of ncRNA: the lncRNA-FLOPAR and miRNA-193. By regulating these ncRNA, SAL mediates a BK channel-dependent apoptosis which, in turn, could be beneficial to prevent vascular senescence induced by microgravity. Working in an aortic smooth muscle cell culture line, the authors showed the molecular mechanism of SAL, which induces a BK channel-dependent apoptosis. Finally, the authors conclude that this effect could explain the delay in cerebrovascular smooth muscle cell senescence induced by microgravity in rats.

Major Comments:

1-     Figure 1 shows that microgravity in rats produces an increase in basilar artery wall thickness and expression of senescence genes (p53 and p16). Interestingly, the administration of SAL reduces the effect of microgravity on vascular remodeling and senescence. Then the authors repeated the measurement of p53 and p16 mRNA and protein expression in a cell line derived from aortic smooth muscle cells exposed to H2O2 as an in vitro model used throughout the manuscript.

a)     This reviewer understands that the aorta is a large capacitance vessel that differs in many pathological conditions from small vessels from the central nervous system. How certain is it to extrapolate conclusions obtained in this model to the cerebral vasculature in vivo?

b)     In this figure, the authors induced a senescence phenotype in the aortic cell line by adding H2O2 and showed that SAL reverts this transformation. Conversely, in the rest of the manuscript, the authors used the unmodified cell line (without H2O2 incubation). Thus, subsequent experiments were done in a model that did not show the changes in vascular cell senescence induced by microgravity. The results should be interpreted in this context.

2-     The authors hypothesize that microgravity-induced vascular senescence could be related to apoptosis resistance related to BK channel activity. Figure 2 shows that BK channel and apoptotic proteins expression is increased in cerebral arteries from the microgravity model, and this effect is increased with SAL administration. This means that microgravity per-se induces apoptosis, and SAL potentiates this effect. It is unlikely that these results can explain the changes in vascular hypertrophy and senescence shown in Figure 1 where SAL administration reverts the effect of microgravity.

3-     The expression of the BK channel and apoptotic proteins was unmodified upon administration of SAL in the control rats, so the compound needs a microgravity stimulus to induce BK-dependent apoptosis. However, in Figure 2 F-J the authors showed that SAL induces apoptosis and increases the BK channel and apoptotic proteins expression in the cell line without any treatment. This reviewer understands that there is no correlation between the in vivo and the in vitro models.

4-     The WB in Figure 1D and H (p53) are not representative of the mean data shown in the same figure.

5-     Figure 2H: Did the author obtain only one replicate in the flow cytometry experiment?

6-     Figure 3 A-C is illegible

Reviewer 2 Report

The authors suggest a novel non-coding RNA modulator in the form of salidroside(SAL) which regulates lncrna-FLORPAR sponging miR pathways which significantly promotes apoptosis pathways. This approach is a novel mechanism in cardiovascular biology/signaling. 

While reviewing all the figures and sponging miRs seems to be fairly deployed by the authors, but I have some reservations on this. The in silico predictions containing the putative binding sites of miR-193 was predicted following which luciferase assay was done suggesting cytoplasmic FLORPAR sponging the miR

Why would the authors use the word "presumptive" binding

The manuscript could start with a rationale which is missing 

A pictorial methodology is necessary 

The 17 materials and methods points are not constructive, for example, some of them could be rewritten 

I don't see volcano plots as mentioned by the authors. Pl check 

Looks good!

Reviewer 3 Report

The manuscript presents interesting data on the beneficial effects of salidroside on the cardiovascular system under simulated microgravity conditions, in particular a decrease in the expressions of senescent biomarkers, and an increase in the expressions of BK-dependent apoptotic genes, including large conductance calcium-activated K+ channel (BK), Bax, Bcl-2, and cleaved caspase-3 in vascular smooth muscle cells (VSMCs) in vivo and in vitro. Moreover, the obtained results indicate that SAL is a novel non-coding RNA modulator for regulating LncRNA-FLORPAR sponging miR-193 pathway, which significantly promotes the BK-dependent apoptosis and delayed the cerebrovascular aging-like remodeling during simulated microgravity exposure.

General remark:

The authors should discuss the limitations of the microgravity model applied. This model is ingenious and cheap but if it is a reasonable microgravity model is somewhat doubtful. It creates microgravity conditions for the hindlimbs but does not completely eliminate the hydrostatic gradients, only changing their directions and magnitude. It does not impose microgravity conditions on the cardiovascular system, increasing it instead and changing its direction. The model is closer to immobilization conditions, which, however, is also employed sometimes to model the effects of a space flight.

Remarks:

Abstract and keywords OK.

Introduction: sufficient and appropriate.

Results: Generally properly presented

Page 6: Please remove the word “Tables”

3.5 (p. 13): “These results indicated that miR-193 inhibited apoptosis by a”, this sentence is truncated

Fig. 1: There is nothing to be seen in Fig. 1A. Perhaps this is a question of pdf, please make sure that the original Figure is well visible. Fig. 1F,G, there is an inconsistency between the plots add the legend: the concentration range of 25-1000 μM, and not 25-500 μM as stated in the legend. Fig. 1 H,I: what was the salidroside concentration used?

Discussion: competent but please address the General remark.

Material and methods: Generally well written.

The authors could report the permit number for this animal experiment (if relevant).

P. 21: “ Sprague Dawley rats (weight: 210-220 g)”, information somewhat imprecise; the initial weight of the SUS+SAL group is 222 g on the average

References 42-44: Please report the journals. Please format References according to journal requirements.

Round 2

Reviewer 1 Report

The authors have improved the presentation of the data. In addition, they have limited the discussion to the data obtained. They have corrected most of my observations satisfactorily.

Author Response

Thanks for your comments and helps in improving our work.

Reviewer 2 Report

The manuscript is improved consistently but still some changes ar eneeded

Page 5

Pl correct Weather to 'whether"

Page 21

Microgravity is mis-spelt

Page 22

Pl avoid the word,  "honestly"

Pl correct inconsistence to inconsistency 

Page 24

simulated is mis-spelt 
